# Insurance claims data: a possible solution for a national sports injury surveillance system? An evaluation of data information against ASIDD and consensus statements on sports injury surveillance

Malin Åman,[1] Magnus Forssblad,[2] Karin Henriksson-Larsén[1]

**To cite:** Åman M, Forssblad M, Henriksson-Larsén K. Insurance claims data: a possible solution for a national sports injury surveillance system? An evaluation of data information against ASIDD and consensus statements on sports injury surveillance. *BMJ Open* 2014;**4**:e005056. doi:10.1136/bmjopen-2014-005056

[1]GIH The Swedish School of Sport and Health Sciences, Stockholm, Sweden
[2]Department of Molecular Medicine and Surgery, Karolinska Institut, Stockholm Sports Trauma Research Center, Capio Artro Clinic, Stockholm, Sweden

**Correspondence to**
Malin Åman;
malin.aman@gih.se

## ABSTRACT

**Background:** Before preventive actions can be suggested for sports injuries at the national level, a solid surveillance system is required in order to study their epidemiology, risk factors and mechanisms. There are guidelines for sports injury data collection and classifications in the literature for that purpose. In Sweden, 90% of all athletes (57/70 sports federations) are insured with the same insurance company and data from their database could be a foundation for studies on acute sports injuries at the national level.

**Objective:** To evaluate the usefulness of sports injury insurance claims data in sports injury surveillance at the national level.

**Method:** A database with 27 947 injuries was exported to an Excel file. Access to the corresponding text files was also obtained. Data were reviewed on available information, missing information and dropouts. Comparison with ASIDD (Australian Sports Injury Data Dictionary) and existing consensus statements in the literature (football (soccer), rugby union, tennis, cricket and thoroughbred horse racing) was performed in a structured manner.

**Result:** Comparison with ASIDD showed that 93% of the suggested data items were present in the database to at least some extent. Compliance with the consensus statements was generally high (13/18). Almost all claims (83%) contained text information concerning the injury.

**Conclusions:** Relatively high-quality sports injury data can be obtained from a specific insurance company at the national level in Sweden. The database has the potential to be a solid base for research on acute sports injuries in different sports at the national level.

## INTRODUCTION

Since approximately 18–30% of all acute injuries are sports-related,[1 2] and almost half of these injuries are related to large team

sports such as football, handball and basketball,[2–4] preventive actions are warranted. Prevention of acute sports injuries requires four stages: (1) identifying the magnitude of the problem (epidemiology), (2) identifying risk factors that may contribute to injuries (aetiology), (3) introducing a preventive measure (prevention) and (4) assessing prevention effectiveness by repeating step 1.[5] To obtain comparable data there are guidelines for standardisation of injury definitions and data collection in the literature.[5–13] In football, rugby union, tennis, cricket and thoroughbred horse racing there are consensus statements on these procedures.[6 10–13]

### Strengths and limitations of this study

- The data assessed include a lot of material from a database that covers more than 80% of the sports federations in Sweden at the national level.
- The study is a structured examination of the reliability and validity of data, using standardised guidelines from the sports injury literature.
- A limitation is that the data has not been evaluated for recording rate reliability since there is no national record of sports injuries in the country.

Most countries do not have a national sports injury surveillance system where injury information from several sports is found in the same database, hence direct comparisons between sports are difficult to perform. Data on sports injury epidemiology may instead be obtained from emergency departments,[2 14] insurance companies,[15–18] other records for specific sports[19 20] or from a specific competition.[8 21 22] Studies have found that data from insurance companies could be eligible

in sports injury research.[16 23 24] Finch evaluated data from three insurance companies in Australia, compared these data with the desirable and necessary information suggested by ASIDD (Australian Sports Injury Data Dictionary[25]), and found that relatively high-quality sports and recreational injury data can be obtained from insurance claims.[23]

At the time of the present study, 57 of 70 sports federations under The Swedish Sports Confederation (RF) use the same Swedish insurance company for injury insurance including all organised athletes in each sports federation. This type of insurance covers the athlete for costs associated with their care and treatment after an accidental injury. If the injury results in a medical disability the athlete receives a lump sum in relation to the severity of the disability. In the case of death, a lump sum is paid to the family. All injuries and payments are registered in a database which covers 90% of all athletes in Sweden, at all ages and all levels of sport. Since previous studies have shown that data from insurance companies can be useful in sports injury research,[16 23 26] the aim of this study was to evaluate the use of the database from this particular Swedish insurance company in injury preventive research, and to compare the obtained information with ASIDD and consensus statements on sports injury surveillance.

### Specific aims

The specific aims of this study were to measure the reliability of the sports injury data by reviewing the injury data collection procedures and a sample of injuries within the database. Validity of the data was examined by comparing the included variables within the database with variables in ASIDD and consensus statements from the literature on injury definitions and data collection procedures.[6 10–13 23 25] After these evaluations, improvements to the database will be suggested.

### METHODS

A sports injury can be registered with the Swedish insurance company within 3 years from the time the accident occurred. The definition of an injury eligible for an insurance claim is a new accidental injury related to sports activities, training or competition organised by the sports federation or sports club that is reported to the insurance company. The cause of the injury should be a sudden external force and no overuse injuries are registered. Information regarding the injury and claim is routinely kept in a systematic and structured manner within the sports injury database and is updated over time as new information is available. The majority of diagnoses are made by doctors. From the database, details can be transferred into an Excel file for statistical analysis. The insurance company performs their own audit of the administrators' claim processing. Randomly selected injury claims, a percentage of the total number of claims, are reviewed by senior administrators every

month. Once a year the company auditors also undertake an extensive review of randomly selected injury claims. The latest audit on sports injury claims was performed in the autumn of 2011, and concluded that claim settlements were generally performed with high quality and accuracy.

Randomly selected settled injury claims between 2006 and 2010 including 27 947 unique injuries were exported to an Excel file, and access to a local database for text information surrounding the injury was received from the insurance company. Data included information on sports activity, age, gender, residence at time of injury, date of injury, diagnosis code (type of injury, injured body region and body part, left or right side of body) and type of financial compensation (medical treatment, dental treatment, permanent disability or death). The data also contained information on disability assessment, the administrator and costs. The local database contained text files with descriptions of injury mechanisms, training or competition, surface and environment and protective equipment. It also included information on the administrator's procedure.

To ensure data reliability, the database was reviewed for available and missing information based on desirable and necessary information previously defined in sports injury research.[6 10–13 25] From the 27 947 unique injuries, 310 random samples were chosen to evaluate data consistency and lack of information within the Excel file compared with the text files in the local database. The insurance company's claim and documentation procedures were studied. All administrators, all sports and all ages of athletes were represented in the randomly selected samples. Data validity was examined by comparing the injury data with ASIDD guidelines in order to make comparisons between different sports,[25] and it was also examined relative to consensus statements in specific sports.[6 10–13]

The Australian Sports Injury Data Dictionary, ASIDD (1997), provides standardised guidelines for injury data collection.[25] According to ASIDD, 'Core' items (category 1) should be presented in all sports injury data collections, 'Strongly recommended' items (category 2) should be included, where possible, to give detailed additional injury information and 'Recommended' items (category 3) can provide further data pertaining to injury circumstances (table 1).[23] Comparison of the Swedish insurance company data with ASIDD was performed, in the same way as earlier performed by Finch in Australia.[23] The presentation of items was registered as 'yes' (present in the database), 'partially' (partially present in the database) and 'no' (not present in the database). A scoring system was used to assess the amount of information in the insurance claims according to ASIDD guidelines:

*1* Data item fully present (should be fully coded according to ASIDD guidelines).

*0* Data item totally absent (in ASIDD but not in the insurance database).

*0.5* Data item partially present (some but not all of the details specified in ASIDD were in the database).

**Table 1** Comparison of the insurance database information with the ASIDD recommendations

| | In the insurance data files |
|---|---|
| ASIDD 'Core' data item | |
| Date of injury | Yes |
| Age | Yes |
| Gender | Yes |
| Activity when injured—broad areas | Yes |
| Mechanism of injury | Yes |
| Body region and body chart | Yes |
| Nature of injury | Yes |
| Category agreement score (maximum score = 7) | 7 (100%) |
| ASIDD 'Strongly recommended' data item | |
| Person recording case information | Yes |
| Immediate source of injury record | Yes |
| Area of usual residence | Yes |
| Name of injury place—text | Yes |
| Place of injury—type | Yes |
| Sport and recreation places—specific | Yes |
| Activity when injured—name of sport or activity | Yes |
| Injury factors | Yes |
| Equipment used with intent to protect against injury | Partially |
| Narrative of mechanism of injury | Yes |
| Date of presentation | Yes |
| Treatment | Partially |
| Advice given to injured person | Partially |
| Referral | Partially |
| Treating person | Yes |
| Category agreement score (maximum score = 15) | 13 (87%) |
| ASIDD 'Recommended' data item | |
| Time of injury | No |
| Date of injury record | Yes |
| Part of specific injury place | Partially |
| Phase or aspect of involvement in activity | Partially |
| Grade/level of play | Partially |
| Specific structure injured | Partially |
| Time of presentation | No |
| Reason for presentation | Yes |
| Category agreement score (maximum score = 8) | 4 (50%) |

In order to ensure the validity of the data for specific sports, a comparison with five existing consensus statements on injury definitions and data collection procedures in the literature[6 10–13] was performed.

## Statistics

The computer programs Microsoft Office Excel 2007 and SPSS (SPSS for Windows, V.21.0) were used. Reliability of the claims and documentation procedures are expressed as percentages by dividing the number of missing values with the number of reviewed posts. When validating the insurance data with ASIDD, a data category agreement score was calculated as the sum of the individual scored items. An overall agreement score was calculated as the sum of the three data category agreements to determine the general quality of information.[23] Validation of the data using consensus statements was based on the established information criteria published in the scientific literature for football, rugby union, tennis, cricket and thoroughbred horse racing.[6 10–13]

## Ethical approval

The project is a part of a more extensive investigation which has been approved by the Regional Ethical Committee in Stockholm (Dnr 2012/1436-31/1). Data was decoded from identification numbers, and only the research leaders have access to the data. No individual consent of the athletes has been requested as results will be presented on a group level and the insurance policy also informs the policyholders that data can be used in research.

## RESULTS
### Loss of data
#### Excel file

Lack of a valid social security number occurred in 103 of the 27 947 posts (0.4%) in the Excel file. This was due to foreign athletes practicing their sport in Sweden without a Swedish social security number. The gender and age of these athletes can be determined from the

Åman M, Forssblad M, Henriksson-Larsén K. *BMJ Open* 2014;**4**:e005056. doi:10.1136/bmjopen-2014-005056

text files in the local database. Fifty-five of the 27 947 injuries (0.2%) lacked a diagnosis code but all codes could be identified when reading the texts in the database. Residence at the time of injury (postal code) was missing in approximately 23% of the posts.

### Local database (text files)

Text files in the local database were available for more than 80% of the 310 random samples taken from the total number of claims. From 2011 onwards, text files were presented in almost all the claims. In some claims from 2007 and earlier, the texts were not always in the local database but could be ordered in hard copy format, which are not included in this study. The diagnosis codes were consistent in more than 88% of the samples. Incorrect diagnoses were found in three samples (1%), unknown diagnoses in six samples (2%) and correct location but incorrect injury type in 13 samples (4%).

An evaluation of different disciplines within the sports federation was also performed. For example, the Swedish Ski Association has different disciplines within the sport, and in the database there are different category numbers for alpine skiing, cross-country skiing, snowboarding etc. The Swedish Mixed Martial Arts Federation and the Swedish Motorsport Federation also have different disciplines within the sport, and thus different category numbers in the database. All samples tested were found to be in the correct sport, however consistency between the category number in the Excel file and the discipline described in the text file in the local database was approximately 83%.

### Compliance to ASIDD

Comparison with ASIDD showed that 93% (ie, 28 of 30) of the suggested data items were present to at least some extent. The overall agreement score was 24 of a highest possible score of 30 (table 1). Within 'Core' ASIDD items, no missing data were found. All 'strongly recommended' items were present in some form but full information was missing in four of them (table 1). In 'recommended' items, five of eight were present but three of them did not include full information (table 1). Part of the information regarding ASIDD items was found in the text files within the local database and not in the Excel file.

### Compliance with consensus statements

Compliance with required information in the consensus statements, published in sports-related scientific literature, was high (13 of 18 items) except for body mass, height, dominant leg/arm, position of play and date of returning to sport after injury (table 2). Some differences in desirable information were found among the different sports federations. Information from seven items existed in the insurance database in some but not all claims (table 2).

### DISCUSSION

Most countries do not have a national sports injury surveillance system where sports injury information from several sports is found in the same database, but in the USA there are national sports injury surveillance systems for adolescent high school and collegiate athletes.[27] [28] Thus, insurance data is a possible way of obtaining

**Table 2** Comparison between the information from the insurance database with required information in the consensus statements

| Possible items | Football | Rugby | Tennis | Cricket | Thoroughbred horse racing | Insurance data |
|---|---|---|---|---|---|---|
| Age | x | x | x | x | x | x |
| Gender | x | x | x | x | x | x |
| Body mass | x | x | x | | x | |
| Height | x | x | x | | x | |
| Dominant leg/arm | x | | x | | | |
| Standard/level of play | | x | x | | x | x− |
| Position in play | x | x | | x | | |
| Date of onset | x | x | x | x | x | x |
| Injury diagnosis/nature of injury | x | x | x | x | x | x |
| Injured side | x | x | x | x | x | x |
| Match/training | x | x | x | x | x | x− |
| Circumstances surrounding the injury | x | x | x | x | x | x− |
| Date of return to sport | x | x | x | x | x | |
| Mechanisms | x | x | x | x | x | x− |
| Free text (around the injury) | x | | x | | x | x− |
| Protective equipment | | | | | x | x− |
| Surface (playing ground) | | | x | | x | x− |
| Treatment (medical) | | | | x | x | x− |

x−, data exist in some but not in all claims.

sports injury information.[23] This study confirms that relatively high quality sports injury data can be obtained from the sports injury database of a specific insurance company in Sweden. The database included all athletes of all ages and at different training and competing levels, in 57 sports federations.

In sports injury research, epidemiology and aetiology are essential as the foundation for sports injury prevention.[5] It is well known that variations in definitions and methodologies create differences in the results and conclusions from studies of sports injuries.[5 26 29] It could therefore be difficult to compare results from different studies. The database in this study includes all licensed and unlicensed athletes in 57 sports federations and the definitions and data collection procedures are the same regardless of sport, thus comparisons between sports, gender and ages are easily made. Since the database only includes accidental injuries and no data of overuse injuries is available, it will not cover the whole injury panorama in each sport.

Earlier studies have shown that insurance claims have the potential to provide detailed information about sports injuries; the records are routinely kept in a systematic manner, and comparisons between data can easily be made.[16 23 26] Different studies on sports injuries have used insurance data for different sports, such as netball, handball, football and rugby.[16–18 24] One smaller study in Finland compared injury profiles in football, ice hockey, volleyball, basketball, judo and karate using national insurance register data.[30] Insurance databases have historically been invaluable for guiding road and workplace safety.[31–33] A recent published study on administrative databases, including insurance databases, concluded that they could be a robust research tool. They allow for longitudinal designs, incidence calculation and large sample size and power, even for rare events.[26]

The insurance data in the present study included many of the ASIDD items; all 'core' and 'strongly recommended' items and more than 60% of the 'recommended' items were present.[25] Data items that, according to ASIDD, could be improved related to treatment, referral and advice given to the athlete when injured.[25] Precise and specific information about injury location, phase or aspect of activity, equipment used and specific structure injured was also missing.[25] One detail not registered was the grade or level of play, although it could occasionally be read in the text file that the athlete competed in a national championship.

The results from comparing the injury claim data with desirable information in the consensus statements[6 10–13] show that the insurance database could be useful in research on sports injuries. However, missing data on personal information such as height, body mass, dominant leg/arm, position of play and return to play are important in some research. Information about return to play is often used to calculate the absence of sporting activity and hence the severity of the injury.[5] In the insurance data, injury type and level of permanent medical disability are two methods of grading injury severity. The definition of medical disability is set by 'Insurance Sweden' (the industry organisation for insurance companies in Sweden). http://www.svenskforsakring.se/Global/Invaliditetsintyg/Medicinsk_invaliditet_skador120911.pdf.

A recently published consensus statement for injury surveillance in athletics is based on previous consensus statements[6 10–13] on details of athlete baseline and recordable incident/injury information. It also reflects more profound information on the specific issues facing athletics at the international and domestic levels.[34]

Prospective data collection procedure is considered to be the most appropriate method for sports injury surveillance.[6 10–13] When using insurance claims data, retrospective and prospective study design can be used depending on the starting point of the study. Although some of the claims may be reported to the insurance company retrospectively since the athlete can report an injury up to 3 years after the accident, in rare cases an injury may be reported to the insurance company at the time when the athlete gains knowledge about it (up to 10 years after injury). In the present study where the data was retrospectively analysed, injuries in the database were reported in median 16 days from the day they occurred.

Van Mechelen suggests that an adequate sports injury surveillance system should be sensitive enough to answer 'How many, how often, how long, and serious?'.[35] If these questions are to be answered on a national level, where the magnitude of sports injury problems are investigated by comparing incidence rates between various sports, or detecting trends and changes over time, a system that can detect large numbers of cases is needed.[35] The system should be as simple and unambiguous as possible and should at least contain a universally applicable definition of sports injury, of sports participation and consensus on population at risk and time at risk.[35] The insurance data in this study are contained in a national register which covers all athletes in more than 80% of all sport federations and includes sports with many athletes such as football, ice hockey, handball, floorball, golf, martial arts and motor sports.

Lack of full information within some items compared with ASIDD and the consensus statements suggests that the insurance data could be improved on in the future with the collection of additional details and standardisation of data variables. Some of these changes have been implemented in the present insurance database as a result of this study. For example, by using headlines in the database, the terminology regarding injury mechanisms and injury circumstances, match/training, surface and free text around the injury has been standardised based on desirable sports injury information in the literature.[6 10–13 25] This enhances the possibility for future prospective studies of purpose-design surveillance at the national level with a large number of athletes and a long follow-up time.

Missing desirable information in the insurance data, including total number of athletes in each sport and exposure to sport could be a weakness when performing

epidemiological studies on sports injuries.[5 36] In further studies using data from the insurance company, assessment of sports injury incidence rate (the number of new sports injuries/population at risk over time) or sports injury prevalence (the proportion of injured athletes in the population at risk at any one time) may be calculated by using information from the specific sports federation regarding the number of participants and licensed athletes in each sport.[11 35 37] When conducting studies on exposure-related incidence of injury an estimation of exposure to sport must be performed, for example, by using estimated information from the specific sports federations regarding time spent on training and competition for each age group and standard/level of sport.

Despite the quality of the sports injury surveillance system, one will always be confronted with some sort of bias, recall bias, overestimation or underestimation of sports participation, incomplete responses, non-response, invalid injury description, etc.[35] The insurance data collection is structured and routinely kept in a systematic manner, and the insurance company works to keep the data accurate and truthful. The database population consists of athletes who *report* an injury to the insurance company and therefore the data it contains likely underestimates the number of injuries. A study from 2005 on anterior cruciate ligament injuries in football, floorball, handball and ice hockey used the same insurance data as the present study and compared it to two medical databases to determine recording rate reliability.[18] The result showed that 74% of all injuries sustained were recorded in the insurance database.[18] Roos et al[38] also studying the same database estimated that 98% of the football injuries were registered within 7 years. The amount of financial compensation for each claim from the insurance company may influence the rate of injury registration. More severe injuries that require medical treatment or may result in a permanent medical disability are more likely to be reported, as well as dental injuries where treatment is expensive. An athlete can report an injury within 3 years after the accident, and medical disabilities are assessed in 1–2 years after the accident occurred. This may result in a delay of recording in the database which must be considered when analysing the data.[38]

Despite these weaknesses, the data are important for the detection of serious injuries and injury mechanisms in different sports at the national level. Injury data, such as incidence and severity of injury can be compared between sports, gender and ages and be followed over time, for example, in longitudinal studies of injury prevention measures at the national level.

## CONCLUSION

Before preventative actions can be suggested, sports injury epidemiology, risk factors and injury mechanisms need to be compiled. Since a comprehensive national database from a Swedish insurance company contains most of the desirable information in sports injury

research, and further improvements to the database information have been implemented, this database now has the potential to be a base for research on acute (accidental) injuries in organised sports at the national level. It could also constitute a foundation for assessment of sports injury prevention at the national level.

**Acknowledgements** The authors would like to thank Folksam insurance company, especially Lars-Inge Svensson and Lena Lindqvist is greatly appreciated.

**Contributors** MA has been responsible for the investigation and compilation of the results, and has in close cooperation with the last writer analysed the results and wrote the manuscript. The other author has contributed valuable intellectual comments and revised the manuscript. MA, MF and KH-L have all substantially contributed to the research and to the final document.

**Funding** The Swedish School of Sport and Health Scienses in collaboration with Folksam, a Swedish insurance company.

**Competing interests** None.

**Provenance and peer review** Not commissioned; externally peer reviewed.

**Data sharing statement** No additional data are available.

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
