## [Reviewer comments · BMJ Open]

Some articles will have been accepted based in part or entirely on reviews undertaken for other BMJ Group journals. These will be reproduced where possible.

ARTICLE DETAILS

TITLE (PROVISIONAL)	Insurance claims data - a possible solution for a national sports injury surveillance system? – An evaluation of data information against ASIDD and Consensus statements on sports injury surveillance.
AUTHORS	Åman, Malin; Forssblad, Magnus; Henriksson-Larsén, Karin

VERSION 1 - REVIEW

REVIEWER	Lucy Hammond Senior Lecturer, Department of Physical Therapy and Rehabilitation, University of Bedfordshire, UK
REVIEW RETURNED	19-Mar-2014

GENERAL COMMENTS	It is initially quite unclear from the title and abstract that this is essentially a methodological paper and not one which examines the data from the insurance database to give injury rates While no participants are involved in the study and therefore ethical approval may not be required, there are ethical issues associated with secondary analysis that have not been commented upon. E.g. I would like to see that the insurance data were not identifiable to individuals, are individuals aware that their data might be used for research, etc Many references cited are about 10 years old, which is probably because this paper deals with insurance data which is not now considered to be the optimal way to conduct injury surveillance. Most up to date references would be based on prospective surveillance, which may explain their absence from this paper. The discussion lacks an overall appreciation of some of the wider issues of injury surveillance e.g. around definitions, and does not address the challenge of making comparisons with other studies that have used different definitions. I would like to see greater discussion of limitations around retrospective data collection and whether the authors feel that overuse injuries would be represented in this database. Overall I have concerns over what this paper adds to our knowledge of injury surveillance and feel that the authors should argue this more strongly. This paper presents an analysis of the usefulness of an insurance database in Sweden for conducting injury surveillance based on how complete the fields of data are within the database. The database holds a large number of cases collected over several years which is a strength as achieving statistical power can be problematic. I do
--

however have some issues with the manuscript in its current form.

Principally, I have concerns over what this paper adds to our knowledge of injury surveillance. Insurance data is by its nature retrospective and there are some obvious biases in using it as a source, some of which have been identified by the authors. Prospective data collection is considered to be the most appropriate method for surveillance and in recent years much attention has been directed towards refining definitions and procedures for data collection. The authors have cited many older papers (pre-consensus statements) and I wonder if this is because insurance data are not used so frequently for injury surveillance in sport now. I think it would help to be clear whether this is aimed at community level sport/non-professional sport/school sport/elite level sport for example, so the reader has a clearer idea of how it could be used. Overall, the authors need to argue more strongly the need for this paper - what does it add? Why now? Who will this knowledge be useful to?

ASIDD (1997) is used as the 'gold standard' (page 9, line 40) to measure this database against. Please justify the use of this as the gold standard when other systems such as the consensus statements have been published more recently and cited more widely.

This paper does not really address the wider issues of surveillance around the complex interaction between definitions selected for use in a study as definitions are set by the insurance company and no return to sport date is available. Therefore there will be issues in comparing any data obtained from using this database compared to prospective purpose-designed surveillance. Please comment on this. Please also comment on the threat to capturing overuse injury or reinjury/recurrent injury through using this approach.

As the study does not have human subjects, ethics are not addressed in this paper. However I would like to see some comment around whether individuals were identifiable from the database, or whether individuals ever consented to third party use of the database - what safeguards are in place for personal data?

Can the authors describe who makes the diagnosis, and whether a diagnosis is updated over time as new information comes to light e.g. a scan?

Several statements are made about what sports/athletes are included in the database, but please comment also on what/who is missing from the database.

The paper title and first half of the abstract appear to suggest that the paper will examine the data in the database (eg to produce injury rates) and it later becomes clear it is considering the database itself. Please be clearer in the description around this.

Minor issues:

Page 3 line 5 change cowers to covers, line 11 data are... Not data is

Page 4 line 18 insert union after rugby to distinguish from league

Page 5 line 58 what are data drop outs? Please explain

Page 7 line 38 please elaborate on the categories as it is unclear what is being referred to here

	Table 1 - could the data in the 2nd column be presented as full, partial or absent (or equivalent) rather than yes/no when there are 3 categories? It would be useful also to express the category agreement score as a percentage as well as actual value Reference 3 - separate statistics and for
--	---

REVIEWER	Jerker Sandelin Orton Orthopaedic Hospital, Helsinki, Finland
REVIEW RETURNED	23-Mar-2014

GENERAL COMMENTS	To collect adequate information regarding sports injuries is difficult because of the multitude of different ways information is collected. Emergency clinic sports injury materials differ significantly from injuries treated by team doctors and no definite conclusions can be drawn regarding sports injuries at large out of these studies. Sports injuries reported to insurance companies is another way of collecting data. However, like in my country sports federations have agreements with different insurance companies making collection of sports injuries on a large basis difficult. In this respect is this study where more than 80% of sports federations injuries are reported to the same insurance company valuable. However, these reported sports injuries are in a way selected because they still represent only those sportsmen and women who have an insurance and take part in sports arranged by the different sports federations. With such a high coverage (57 out of 70 federations) this is perhaps as close one can get in trying to obtain valuable and correct data concerning sports injuries. The number of injuries seems high but spread out on 8 years gives some 3500 injuries pro year. This number is not high compare to the annual number of sports injuries expected to happen in Sweden annually The information obtained from the database seems however, to be sufficient (good compliance with ASIDD) that problems of sports injuries and preventive measurements can be undertaken. The definition of sports injury should be clarified. What is meant by receiving medical attention? Are these injuries all seen by doctors or can these injuries be reported by physiotherapists? Other medical persons?
--

VERSION 1 – AUTHOR RESPONSE

Reviewer 1:

Q1: It is initially quite unclear from the title and abstract that this is essentially a methodological paper....;

Response: The title has now been changed.

Q2: I would like to see that the insurance data were not identifiable to individuals, are individuals aware that their data might be used for research, etc....

Response: This has now been clarified in the manuscript.

Q3: Many references cited are about 10 years old....

Response: The reference list has now been updated.

Q4: The discussion lacks an overall appreciation of some of the wider issues of injury surveillance e.g. around definitions, and does not address the challenge of making comparisons with other studies that have used different definitions.

Response: The discussion of these issues has now been extended.

Q5: I would like to see greater discussion of limitations around retrospective data collection

Response: The data in the database was not collected retrospectively, though the withdrawal of data from the database was. Thus there was no limitation in data collection other than that already existing data was the only data that could be withdrawn from the database. This has now been clarified in the manuscript.

Q6: and whether the authors feel that overuse injuries would be represented in this database.

Response: Of course overuse injuries are of a great interest in sports though the scope of this paper is a methodological study on the use of an insurance database as a national register for acute traumatic sports injuries. The preventive measures taken for overuse injuries and traumatic injuries most often are different. In this planned series of studies we focus on the traumatic injuries and their prevention.

Q7: Overall I have concerns over what this paper adds to our knowledge of injury surveillance and feel that the authors should argue this more strongly.

Response: The novelty of this paper is that there is one database with sports injury insurance claims for a whole country and 90% of all athletes, both licensed and unlicensed, is covered by this insurance company.

Q8: Who will this knowledge be useful to?

Response: The knowledge will be used to design preventive actions in different sports, for different age groups and for different genders, and to follow the implementations of these preventive actions over time on a national level.

Q9: ASIDD (1997) is used as the 'gold standard' (page 9, line 40)

Response: This line has been rephrased.

Q10: ... no return to sport date is available.

Response: In this study we are also interested in the consequences of the sports injuries outside sports. Medical disability is thus a precise way to define severity of an injury.

Q11: Therefore there will be issues in comparing any data obtained from using this database compared to prospective purpose-designed surveillance.

Response: This is described in the result section and has been elaborated in the discussion.

Q12: Can the authors describe who makes the diagnosis, and whether a diagnosis is updated over time as new information comes to light e.g. a scan?

Response: This is now clarified in the method.

Q13: Several statements are made about what sports/athletes are included in the database, but please comment also on what/who is missing from the database.

Response: Since this a methodological study, data analyses are not performed. We thus did not find it of interest to comment on which sports were/were not included in the database but only to what extent the database covered the different sports and percentage of athletes. If this is of interest we could of course provide this data as an amendment.

Q14: Minor issues:

Response: All these changes have been performed.

Page 3 line 5 change covers to covers, line 11 data are... Not data is

Page 4 line 18 insert union after rugby to distinguish from league

Page 5 line 58 what are data drop outs? Please explain

Page 7 line 38 please elaborate on the categories as it is unclear what is being referred to here

Table 1 - could the data in the 2nd column be presented as full, partial or absent (or equivalent) rather than yes/no when there are 3 categories? It would be useful also to express the category agreement score as a percentage as well as actual value

Reference 3 - separate statistics and for

Reviewer 2:

Q1: The number of injuries seems high but spread out on 8 years gives some 3500 injuries pro year. This number is not high compare to the annual number of sports injuries expected to happen in sweden annually

Response: The withdrawn data was a random sample of almost 28000 injuries out of approximately 90000 injuries during this period.

Q2: The definition of sports injury should be clarified. What is meant by receiving medical attention?

Response: The definition is now clarified in the text.

Q3: Are these injuries all seen by doctors or can these injuries be reported by physiotherapists? Other medical persons?

Response: Medical evaluation of the injuries is required to be performed by a medical doctor. In rare cases an evaluation of a physiotherapist has been approved.

VERSION 2 – REVIEW

REVIEWER	Lucy Hammond University of Bedfordshire, UK
REVIEW RETURNED	26-Apr-2014

GENERAL COMMENTS	A minor issue remains over the clarity of the dichotomous yes/no presentation of results to a question that has 3 coding categories The discussion of the limitations of the study has been strengthened in the revised manuscript, however there it is still weak regarding the handling of the potential retrospective nature of some of the data and the absence of exposure data to estimate exposure related incidence This manuscript has been improved by the amendments made by the authors, and the purpose of the study is now much clearer and elements have been more robustly described. The authors maintain that the data were not obtained retrospectively, however injuries can be reported up to 3 years after they happened and in some cases up to 10 years (page 31 line 20). To my mind, this means that some injuries might have been retrospectively reported some years after they occurred, not prospectively at the time of occurrence. It may be that only a small proportion of the database is affected by this, and it would be interesting to know how many cases this affected. At the least, it should be made clear that this risk of bias is present in the data set.
---

	The authors make several statements about using the database to compare injury incidence between sports, but in the absence of exposure data this would be difficult to perform. I would still maintain that the clarity of Table 1 is impaired by the dichotomous yes/no presentation of results to a question that has 3 coding categories (fully present, totally absent, partially present) and would prefer to see this presented differently, maybe partial in brackets next to those coded as 0.5? The authors may not be aware that another consensus statement has recently been published, for athletics, and may want to consider incorporating this recent publication into their paper if athletics features in their database http://bjism.bmj.com/content/48/7/483.full
--	--

VERSION 2 – AUTHOR RESPONSE

Reviewer comments:

Q1: A minor issue remains over the clarity of the dichotomous yes/no presentation of results to a question that has 3 coding categories

Response: This has now been clarified in the manuscript by using the same terminology as professor C Finch in Australia used in her study where she evaluated data from three insurance companies in Australia using the same method as in the present study.

Q3: ...there it is still weak regarding the handling of the potential retrospective nature of some of the data..

Response: It is now clarified that this is a small proportion of the database that is affected by this, and the medium days of reporting injury (16 days) is stated.

Q4: ...the absence of exposure data to estimate exposure related incidence

Response: The discussion of this issue has now been extended.

Q5: The authors may not be aware that another consensus statement has recently been published, for athletics..

Response: We are aware of this, and was listening to the authors of this publication on the IOC World Conference on Preventing of Injury and Illness In Sport in Monaco in April this year. The definitions and guidelines presented are based on previous consensus statements and reflect also more profound information on the specific issues facing Athletics. This reference is now added to the reference list.